# Early Wound Healing Score (EHS): An Intra- and Inter-Examiner Reliability Study

**DOI:** 10.3390/dj7030086

**Published:** 2019-09-01

**Authors:** Lorenzo Marini, Philipp Sahrmann, Mariana Andrea Rojas, Camilla Cavalcanti, Giorgio Pompa, Piero Papi, Andrea Pilloni

**Affiliations:** 1Section of Periodontics, Department of Oral and Maxillofacial Sciences, “Sapienza” University of Rome, 00161 Rome, Italy; 2Clinic of Preventive Dentistry, Periodontology and Cariology, Center of Dental Medicine, University of Zurich, 8032 Zurich, Switzerland; 3Oral Surgery Unit, Department of Oral and Maxillofacial Sciences, “Sapienza” University of Rome, 00161 Rome, Italy

**Keywords:** classification, diagnosis, gingiva, reproducibility of results, soft tissue injuries, wound healing

## Abstract

The early wound healing score (EHS) was introduced to assess early wound healing of periodontal soft tissues after surgical incision. The purpose of this study is to evaluate the intra- and inter-examiner reliability of the EHS. Six examiners with different levels of training and clinical focus were enrolled. Each examiner was trained on the use of the EHS before starting the study. Thereafter, 63 photographs of three different types of surgical incisions taken at day 1, 3 or 7 post-operatively were independently evaluated according to the proposed assessment method. A two-way random intra-class correlation coefficient (ICC) and 95% confidence interval (CI) were used to analyze the intra- and inter-examiner reliability for the EHS. The inter-examiner reliability for the EHS was 0.828 (95% CI: 0.767–0.881). The intra-examiner reliability ranged between 0.826 (95% CI: 0.728–0.891) and 0.915 (95% CI: 0.856–0.950). The results therefore show an “almost perfect agreement” for intra- and inter-examiner reliability. The EHS provides a system for reproducible repeated ratings for the early healing assessment of incisions of periodontal soft tissues. Even when used by examiners with different clinical experience and specialty, it shows a high correlation coefficient.

## 1. Introduction

Many of the cellular and molecular events in the healing of periodontal soft tissue wounds are similar to those seen in wounds in extra-oral sites, through the stages of haemostasis, inflammation, proliferation and remodeling [1,2]. However, periodontal wounds present special features in many aspects. Firstly, the dento-gingival unit stands out for a mineralized tissue interface at the junction of epithelium and connective tissue [3,4,5]. Secondly, wounds of the oral mucosa demonstrate accelerated healing and reduced scar formation as compared to cutaneous wounds [6,7,8,9,10,11,12].

The cellular response after wounding begins early, showing considerable changes already at 12–24 h post injury [13]. In particular, approximately 24 h after surgery keratinocytes start moving along the wound margins aiming at re-establishing the tissue integrity and allowing further steps of healing in an environment that is protected from microbiological and mechanical impact [14]. From a clinical point of view, healing during the first postoperative days is crucial for the maintenance of wound stability and therefore for successful treatment outcome [4,5,15,16].

Post-surgical monitoring is an important concern in periodontology, since the clinician may soon become aware of possible complications and may react in order to reestablish conditions for better clinical outcomes [17].

In the last 30 years, several qualitative and semi-quantitative assessment methods to evaluate early wound healing of periodontal soft tissues have been proposed and subsequently modified. Each evaluation system showed specific indications but also some limitations. Among the most cited tools developed for the assessment and monitoring of oral soft tissue early wound healing, the Healing Index (HI) by Landry et al. was the first [18]. The HI was intended to assess healing of extraction sockets one week after intervention based on the following parameters: tissue color, bleeding response to palpation, presence of granulation tissue, characteristics of the incision margins and the presence of suppuration. Each wound can be scored from 1 (very poor healing) to 5 (excellent healing). In 2003, Wachtel et al. published the Early Wound Healing Index (EHI) for the evaluation of early wound healing 1–2 weeks after surgical treatment of infra-bony defects [19]. The EHI subdivides the quality of healing into scores. Flap closure is assessed as complete or incomplete, the amount of fibrin quantified and the presence of necrosis recorded. In 2005, Huang et al. presented the Wound Healing Index (WHI) to clinically assess the early healing at 2 weeks following a coronally advanced flap root coverage procedure [20]. The WHI estimates absence or extent of gingival edema, erythema, suppuration, patient discomfort, flap dehiscence and suppuration. For each surgical area a score from 1 (best quality healing) to 3 (worst quality healing) is possible. In 2018, Trombelli et al. published the Gingival Healing Index (GHI), which specifically assessed the post-surgery conditions of the interdental papilla at 7 and 21 days [21]. This composite index, whose GHI ranges from 2 (worst quality of healing) to 6 (optimal quality of healing), is the sum of independently evaluating both the severity of wound dehiscence and the profile of the oral and buccal aspects on a three-step scale. In the same year, Hamzani and Chaushu suggested a novel scale, the IPR Wound Healing Scale, ranging from 0 (worst quality of healing) to 16 (best quality of healing) [22]. This scale, assessed during wound healing after implant placement, scores distinctive parameters of the injured site for each of the three wound aspects, inflammatory (I), proliferation (P) and remodeling (R), assessed at 3–5 days, 14 days and 6 weeks respectively. Furthermore, the Early Wound Healing Score (EHS) was recently introduced by our group to assess wound healing by primary intention in periodontal soft tissues after surgical incisions [23]. The EHS was based on the evaluation of clinical signs of re-epithelialization, hemostasis and inflammation. The sum of the single scores for these 3 parameters calculates the EHS, which ranges between 0 (worst possible wound healing) to 10 points (ideal wound healing). It was intended to be used from day 1 post surgery onward, allowing the most significant events in oral wound healing to be monitored according to the contemporary interpretation of “early” phase by the most relevant scientific sources [24].

The application of EHS shows the following special features: (i) comprehensive inclusion and accurate description of all the clinical manifestations of primary intention early wound healing, properly selected to be measured from day one post-operatively; (ii) subdivision of different manifestations of early wound healing into three sub-parameters of the score, where are related since they are considered as dissimilar clinical presentations of the same biological phase (re-epithelialization, hemostasis and inflammation) rather than being interconnected because they are associated to the same wound healing status (e.g. poor or excellent quality of healing); (iii) distinction of wound healing attributes, comprised in the sub-parameters, between primary and secondary, with a specific effect on the overall evaluation; (iv) independent assessment of each clinical symptom that describes the early wound healing according to the values assigned for the respective sub-parameter; (v) integration of data to obtain a final score; (vi) possibility of its use for all incisional wounds that are supposed to heal by first intention. All these characteristics taken together seem to represent important and practical advantages if compared to previous methods suggested for the same purpose, supporting the general use of the EHS as a potential common standard for communication between researchers and clinicians. However, its clinical practicality in terms of reliability has not been evaluated.

The primary aim of this study is therefore to evaluate intra- and inter-examiner reliability of the EHS among examiners with different levels of education and different clinical focus/specialization. A secondary aim was to determine if different types of surgical incisions and the time interval at which wounds are assessed have any influence on the reliability of the EHS.

## 2. Materials and Methods

Reliability of the early wound healing score (EHS) were tested according to the Guidelines for Reporting Reliability and Agreement Studies (GRRAS) [25].

For this type of study, no ethical committee approval was required by the “Sapienza” University of Rome.

### 2.1. Selection and Preparation of Photographs

Before beginning this study, the primary author (LM) collected post-treatment photographs of surgical incisions from patients that underwent periodontal or implant surgery. Patients had been treated with different surgical procedures by students of the postgraduate program in periodontology at the Department of Oral and Maxillofacial Sciences, “Sapienza” University of Rome, from September 2016 to June 2018.

Standardized digital photographs were taken perpendicularly to the long axis of the teeth comprised in the surgical area. If needed, pictures were resized to show only surgical incisions and their contour area, maintaining the original proportions at a 300 dpi resolution. Neither color, brightness nor contrast adjustments were performed. Only images that depicted the complete surgical site and allowed for safe evaluation were used for the test assessment.

In order to select the adequate number of photographs needed to verify the intra- and inter-examiner reliability of EHS, comparable studies were examined and a preliminary calibration with the examiners was conducted [26,27,28]. After that, the sample size was calculated using a minimal acceptance level of the intra-class correlation coefficient (ICC) for the inter-rater agreement of 0.60, with an alternative hypothesis of 0.85, α = 0.05 and β = 0.05 [29]. On the basis of these parameters, a minimum of 18 images was required for the assessment by six examiners.

The sample included photographs of 3 types of surgical incisions (21 photographs of vertical releasing incisions, 21 of crestal linear incisions and 21 of papilla base incisions). Each group of 21 images, referring to a distinct incision type, comprised 7 photographs taken at 3 different time points of early wound healing (1, 3 and 7 days). Therefore, for each time point there were 21 images as well. The photographs were anonymized regarding patient and operator identity and the nature of the surgical procedure performed.

For assessment, the final composite sample of 63 images was assembled in a slideshow and presentation software. Moreover, a standardized assessment protocol was prepared for data recording.

### 2.2. Selection and Training of Examiners

Six examiners showing different levels of education and clinical focus were selected to contribute in this study. Therefore, a fourth-year pre-doctoral student of DDS (CC), a PhD student in periodontology with 8 years of clinical experience (MAR), a PhD student in implant prosthodontics with 6 years of clinical experience (PP), a specialist in periodontology with 17 years of clinical experience (PS), the director of a postgraduate master’s program in periodontology with 30 years of clinical experience (AP) and the director of a postgraduate master’s program in implant prosthodontics with 30 years of clinical experience (GP) were enrolled for the reliability assessment.

Before assessment, each examiner received training on the nature and applicability of EHS by the primary author (LM). Therefore, five photographs of clinical cases that were not included in the study with their respective EHS values were shown and explained to the participants.

### 2.3. Early Wound Healing Assessment with EHS

The six examiners independently assessed all 63 images according to EHS indications. Assessment was repeated after an interval of 1 week. No time limitation was given to the examiners to assess each photograph.

EHS assessment was composed of the parameters clinical signs of re-epithelization (CSR), haemostasis (CSH) and inflammation (CSI) [23]:CSR = 0 points: visible distance between incision margins, 3 points: contact between incision margins, 6 points: merged incision margins; CSH = 0 points: bleeding at the incision margins, 1 point: presence of fibrin on the incision margins, 2 points: absence of fibrin on the incision margins; CSI = 0 points: redness involving >50% of the incision length and/or pronounced swelling, 1 point: redness involving <50% of the incision length, 2 points: absence of redness along the incision length.

For each parameter, the worst score observable on the pictures was registered. The sum of the scores of these three parameters made the EHS score. Accordingly, the EHS for ideal wound healing was ten points, the worst was zero points. The EHS was scored zero points in the presence of suppuration independently of the ratings for the three single parameters.

After finishing the assessment, each examiner returned the worksheet for further analysis by LM. The early wound healing assessment with EHS of 9 different clinical cases is illustrated in Figure 1.

### 2.4. Statistical Analysis

Descriptive statistics (means and standard deviations) were calculated. Intra- and inter-examiner reliability of the EHS and each single parameter (CSR, CSH, CSI) were calculated using a two-way random intraclass correlation coefficient (ICC) and 95% confidence interval (CI). A respective sub-analysis for the 3 types of incisions and the 3 post-operative time intervals of early wound healing was also performed.

According to the assessment method by Landis and Koch, a six-level nomenclature was used to evaluate the level of reliability: poor agreement = <0.00, slight agreement = 0.00 to 0.20, fair agreement = 0.21 to 0.40, moderate agreement = 0.41 to 0.60, substantial agreement = 0.61 to 0.80, and almost perfect agreement = 0.81 to 1.00 [30]. For data computation a statistical software package (IBM Corp. Released 2017. IBM SPSS Statistics for Macintosh, Version 25.0. Armonk, NY, USA: IBM Corp.) was used.

## 3. Results

The sample included 39 photographs of incisions performed in the anterior area (from canine to canine) and 26 in the posterior area (from the first premolar to the second molar). Forty-five of the incisions were localized in the upper arch and 20 in the lower arch.

The time needed for the pre-study training and calibration of each examiner was 60 min.

All written assessments could be used for statistical analysis. 

Table 1 shows medians and interquartile ranges of EHS of each examiner, time interval and incision type. Minor differences in terms of median EHS values were found among the examiners but did not exceed the score for >1.0. In contrast, major variations were identified between incisions evaluated at 7 days and those assessed earlier. The same was found when comparing vertical releasing incisions with the other two types of incision.

The ICC for the total inter-examiner reliability for the EHS was 0.828 (0.767–0.881), indicating an “almost perfect” agreement according to Landis and Koch [30].

The sub-analysis for the three EHS parameters (Table 2) showed that the parameter that reached the highest level of agreement among all participants was CSR (ICC: 0.777; 95% CI: 0.704–0.843, almost perfect agreement), while the parameter with the lowest agreement was CSH (ICC: 0.619; 95% CI: 0.520–0.718, substantial agreement). CSI resulted in substantial agreement (ICC: 0.694; 95% CI: 0.604–0.778). None of the incisions was scored 0 because of suppuration.

The inter-examiner reliability was sub-analyzed for incision types and post-operative time points (Table 3 and Table 4). Concerning the incision types, the highest level of inter-examiner agreement was recorded for the vertical releasing incisions (ICC: 0.866; 95% CI: 0.773–0.934, almost perfect agreement), while the lowest agreement was found for the crestal linear incision (ICC: 0.791; 95% CI: 0.662–0.893, substantial agreement). With regards to the evaluation time, the inter-examiner agreement was higher when the incisions were scored at 7 days post-operative (ICC: 0.889; 95% CI: 0.810–0.946, almost perfect agreement) compared to 3 days (ICC: 0.759; 95% CI: 0.620–0.875, substantial agreement) and 1 day (ICC: 0.775; 95% CI: 0.642–0.884, substantial agreement).

In order to establish if the results provided by each examiner could be reproducible when reassessed, examiners evaluated the same sample of photographs a second time. For all participants, the total intra-examiner reliability was very high, ranging between 0.826 (95% CI: 0.728–0.891) and 0.915 (95% CI: 0.856–0.950). The ratings from all the involved clinicians are shown in Table 5.

## 4. Discussion

Different scores have been published to assess early wound healing in oral soft tissues [18,19,20,21,22]. The EHS was newly introduced in order to evaluate surgical incision healing by primary intention, from day 1 post-operative onwards, using an adequate definition and a comprehensive inclusion of clinical parameters, allowing an independent assessment [23]. To the best of our knowledge, none of the previously proposed methods, including the EHS, was validated by studies that measured the reproducibility of the repeated assessments obtained by the same examiner and the agreement among different examiners. Since early wound healing assessment must produce results that are consistent over a range of conditions and circumstances, this study aimed at calculating the intra- and inter-examiner reliability of the EHS between examiners showing different levels of education and/or experience in various specialties using photographs of 3 types of surgical incisions taken at 1, 3 or 7 days post-surgery.

Based on the assessment method of Landis and Koch [30], the EHS showed an “almost perfect” level of inter-examiner agreement among all examiners (ICC: 0.828; 95% CI: 0.767–0.881; Table 2). This means that the EHS is valid for the examination of early surgical incision healing by primary intention. In evaluating the results of the present study, one should keep in mind that all examiners received initial training and were therefore well calibrated. Indeed, several studies reported a positive effect of observer training in improving the reliability of examiner evaluation [31,32]. Moreover, no more than 60 min was needed to train the examiners. This seems to be enough time to read the publication of the EHS, where all the necessary information is provided [23].

Sub-parameter CSR showed the highest agreement (ICC: 0.777; 95% CI: 0.704–0.843, substantial agreement; Table 2), leading to a homogeneous comprehension of the points assignment: 6 points in the case of merged incision margins, representing the final stage of the wound re-epithelialization; 3 points in the case of contact between incision margins, indicating the maintenance of primary healing and wound margins approximation; 0 points in the case of distance between incision margins, implying wound dehiscence. The high ICC value reached by the CSR was considered of remarkable importance since wound closure was considered the main early wound healing outcome and this parameter accounted for the 60% of the total score.

CSH turned out to show the worst agreement (ICC: 0.619; 95% CI: 0.520–0.718, substantial agreement; Table 2). Even though these results were positively interpreted, data revealed a superior disagreement in the assessment of some cases. It was clarified that the presence of fibrin should be assessed only when visible between wound margins, because it would be indicative of an imperfect margins approximation or a partial/complete wound dehiscence, and not when it was found in the form of deposits covering the surrounding area of the incision, since in this case it would be considered of no clinical and biological relevance. Fewer doubts were raised among examiners in the case of the presence of bleeding, which was scored 0 points both in the case of spontaneous and triggered bleeding.

Finally, CSI obtained substantial agreement with 0.694 (0.604–0.778; Table 2). This result was interpreted as satisfactory, considering that the clinical assessment of inflammation often bears a high risk of subjectivity. The EHS was designed to reduce this risk by precisely defining the criteria of assessment of this event. Regarding redness, its evaluation was performed on a purely objective quantification of the extent. Concerning the swelling, it was suggested its presence only be considered when obvious, to diminish the risk of a low specificity. Moreover, both manifestations, redness and swelling, should only be considered if they exactly refer to the incision area, which was defined as the area within a 2 mm distance from the incision margins.

The EHS was described in a preliminary study where incisions were evaluated at day 1 post-operatively [23]. The reason was to illustrate the ability of this score to assess wounds at this early time interval, which was considered to be significant because, immediately after the haemostatic and inflammatory phases, re-epithelialization begins approximately 24 h following tissue injury [14]. However, in the present study the authors chose to include, among the 63 photographs, 42 images of surgical incisions taken at day 3 and 7 after surgery, in addition to 21 pictures taken at day 1, to prove the use of the EHS in different periods. Although the EHS could be used in later stages, in order to validate its use, it was decided to focus on an earlier assessment. The use of the present method, 1 to 3 days after intervention, might be primarily in the researchers’ interest since it provides the opportunity—besides other individual and appropriate quantitative measurements—to follow, score and hopefully better understand the clinical behavior of periodontal soft tissues after incision. Regarding its use by clinicians, it could represent a simplified record of early wound healing at a practical point of time like 7 days after an intervention. The inter-examiner reliability demonstrated impeccable agreement for all the three intervals of early wound healing by which wounds were evaluated, certifying the proper application of the EHS at day 1 and after. Interestingly, the highest ICC was obtained for the incisions at the seventh day of healing (ICC: 0.889; 95% CI: 0.810–0.846, almost perfect agreement; Table 3), while the lowest was for the ones at the third day (ICC: 0.759; 95% CI: 0.620–0.875, substantial agreement; Table 3). A different distribution of the EHS values at day 7 compared to those of the incisions evaluated at day 1 and 3 was observed (Table 1). In fact, such wounds exhibited higher EHS values, which indicated that they were in an advanced stage of early healing and could result in an easier assessment by the examiners. However, these optimal healing results are usually obtained at this time interval when respecting indication, such as patient and site selection, proper operation techniques and operator training.

To better standardize the evaluation, in the original article the EHS was assessed only at the level of vertical releasing incisions. However, the use of this score was suggested to evaluate other types of primary intention wound healing. Indeed, the authors comprised in the sample of 63 photographs, 42 images equally subdivided between crestal linear incisions and papilla base incisions, in addition to 21 pictures of vertical releasing incisions, to test the applicability of the EHS in a wider variety of settings. The inter-examiner reliability demonstrated an “almost perfect” agreement for all the three wound types, indicating the correct use of the EHS in all the incisions tested. A lower ICC was obtained for the crestal linear incisions (ICC: 0.791; 95% CI: 0.662–0.893, substantial agreement; Table 4) and the papilla base incisions (ICC: 0.822; 95% CI: 0.709–0.911, almost perfect agreement; Table 4), compared to that of the vertical releasing incisions. (ICC: 0.866; 95% CI: 0.773–0.934, almost perfect agreement; Table 4). A probable hypothesis to support these results could be that the crestal linear incisions and the papilla base incisions exhibited a higher complexity from an anatomical point of view and underwent greater tissue remodeling, leading to a more difficult interpretation of wound healing. Nevertheless, differences in the distribution of the EHS values among these two incision types and the vertical releasing incisions were observed (Table 1).

Concerning the intra-examiner reliability, all six examiners revealed a nearly perfect agreement, ranging between 0.826 (95% C 0.728–0.891; Table 5) and 0.915 (95% CI: 0.856–0.950; Table 5). Even though the lowest agreement was obtained by an examiner with less experience in dental practice, these findings demonstrated a suggestive coherence of the participants to their first evaluation of the EHS when exposed to the same sample of photographs after 1 week. The results could be understood as a consequence of the standardized and concise structure of EHS.

During the course of the study, the authors had the opportunity to examine in detail the application of the EHS and felt the need to report the following observations: (a) the EHS is a description of the early wound healing process, therefore its clinical relevance in predicting the final outcome of any intervention should be determined by future prospective studies; (b) the potential clinical relevance of the EHS in predicting the long term results is specific for each surgical procedure, and consequently scores of different types of incisions should not be compared with this aim; (c) comparisons between scores of different incisions should be accomplished always keeping in mind the time point at which each of them is evaluated, for a correct understanding of the dissimilar clinical manifestations; (d) incisions with the same EHS generated values, at the same post-operative time, by different combinations of the sub-parameters should be considered as wounds with equivalent healing during the early phase; (e) a possible inter-relationship between sub-parameters could be further investigated to determine the usefulness of their independent assessment.

Inter- and intra-examiner agreement stratified by the surgical area where incisions were localized was not calculated. Even though it could biologically affect the outcome of early wound healing, the anatomical site was not considered significant when evaluating the repeatability of assessments using photographs. Conversely, it would have been important—in terms of agreement—if the evaluation had been carried out clinically. In this case, the access to the surgical area and the operator’s visibility could have influenced the proper score of early wound healing. Differences in agreement between dissimilar surgical sites could be taken into account in further investigations.

Limitations of this study were mainly represented by the above-mentioned use of photographs in place of a direct clinical evaluation. Moreover, no control was performed regarding the computer monitor quality and the environment characteristics through which each examiner recorded his or her own assessment. Finally, despite the sufficient power of the study, a larger number of photographs and examiners could be considered for further research.

## 5. Conclusions

In conclusion, the analysis resulted in adequate intra- and inter- examiner reliability when evaluating the primary healing of periodontal soft tissues after different types of surgical incisions when using the EHS. Accordingly, the EHS appears to be an appropriate tool to assess early wound healing for dental professionals and especially researchers.

## Figures and Tables

**Figure 1 dentistry-07-00086-f001:**
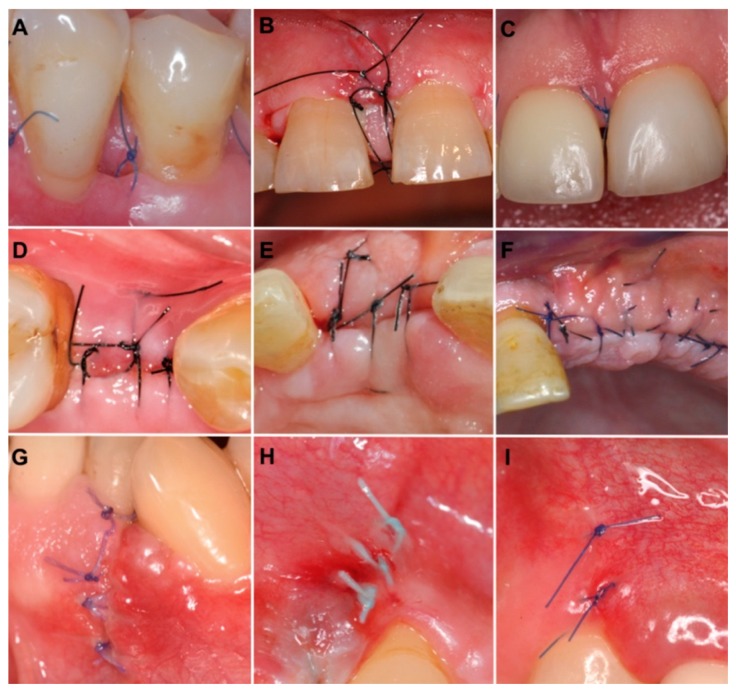
Clinical photographs of surgical incisions evaluated according to early wound healing score (EHS), with corresponding values of clinical signs of re-epithelialization (CSR), haemostasis (CSH) and inflammation (CSI). (**A**) The value of EHS is 4 (CSR:0; CSH:2; CSI:2). (**B**) The value of EHS is 4 (CSR:3; CSH:1; CSI:0). (**C**) The value of EHS is 10 (CSR:6; CSH:2; CSI:2). (**D**) The value of EHS is 4 (CSR:0; CSH:2; CSI:2). (**E**) The value of EHS is 5 (CSR:3; CSH:0; CSI:2). (**F**) The value of EHS is 10 (CSR:6; CSH:2; CSI:2). (**G**) The value of EHS is 4 (CSR:3; CSH:1; CSI:0). (**H**) The value of EHS is 6 (CSR:3; CSH:2; CSI:1). (**I**) The value of EHS is 10 (CSR:6; CSH:2; CSI:2).

**Table 1 dentistry-07-00086-t001:** Median and quartiles of early wound healing score of each examiner, time interval and incision type.

	Median	Lower Quartile	Upper Quartile
Examiner			
DDS student	8	6	10
PhD student in periodontology	8	6	9
PhD student in implant prosthodontics	7	5	9
Specialist in periodontology	7	5	10
Director of master’s program in periodontology	7	5	10
Director of master’s program in implant prosthodontics	8	5	10
Time interval			
1 day	6	5	9
3 days	7	4	9
7 days	10	7	10
Incision type			
Vertical releasing incision	9	7	9.25
Crestal linear incision	6	4	9.25
Papilla base incision	6.5	4	10

**Table 2 dentistry-07-00086-t002:** Inter-rater reliability stratified by EHS sub-score.

EHS Sub-Score	Mean ± SD	ICC (95% CI)
Clinical Signs of Re-epithelialization	4.16 ± 2.15	0.777 (0.704–0.843)
Clinical Signs of Haemostasis	1.53 ± 0.65	0.619 (0.520–0.718)
Clinical Signs of Inflammation	1.33 ± 0.77	0.694 (0.604–0.778)
Total	7.02 ± 2.72	0.828 (0.767–0.881)

**Table 3 dentistry-07-00086-t003:** Inter-rater reliability of early wound healing score stratified by time intervals (T) of early wound healing.

T	Mean ± SD	ICC (95% CI)
1day	6.55 ± 2.22	0.775 (0.642–0.884)
3days	6.54 ± 2.54	0.759 (0.620–0.875)
7days	7.96 ± 3.10	0.889 (0.810–0.946)

**Table 4 dentistry-07-00086-t004:** Inter-rater reliability of early wound healing score stratified by incision types.

Incision Type	Mean ± SD	ICC (95% CI)
Vertical releasing incision	8.23 ± 1.62	0.866 (0.773–0.934)
Crestal linear incision	6.35 ± 2.89	0.791 (0.662–0.893)
Papilla base incision	6.47 ± 3.02	0.822 (0.709–0.911)

**Table 5 dentistry-07-00086-t005:** Intra-rater reliability obtained after application of the early wound healing score.

Examiners	Mean ± SD	ICC (95% CI)
DDS student	7.31 ± 2.58	0.826 (0.728–0.891)
PhD student in periodontology	6.73 ± 2.76	0.915 (0.856–0.950)
PhD student in implant prosthodontics	7.11 ± 2.73	0.910 (0.848–0.946)
Specialist in periodontology	7.15 ± 2.68	0.894 (0.830–0.934)
Director of master’s program in periodontology	7.15 ± 2.66	0.902 (0.843–0.939)
Director of master’s program in implant prosthodontics	7.23 ± 2.90	0.890 (0.825–0.932)

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
