# Peer review of "Early Wound Healing Score (EHS): An Intra- and Inter-Examiner Reliability Study"

_dentistry, 2019, doi:10.3390/dj7030086_

Round 1

Reviewer 1 Report

The authors describe a study analyzing the intra- and inter-examiner reliability of the early wound healing score (EHS). 

An evaluation regarding its reliability has not been published so far. 

Three different incision types performed in periodontal surgery were presented at different points of time (1, 3 and 7 days after surgery) to six examiners. The examiners had to score the healing status according to the EHS score after looking at pictures of different patients. 

The EHS includes an assessment of the re-epithelization, haemostasis and inflammation.

The intra and inter-examiner reliability was very high for this score. 

Three parameters are scored. The re-epithelializytion had the highest agreement. Inflammation had a substantial agreement whereas the haemostasis had the worst agreement among the examiners.

The study was well performed. Unfortunately, the authors used photographs, but this fact is addressed in the discussion section.

Author Response

Point 1: The study was well performed. Unfortunately, the authors used photographs, but this fact is addressed in the discussion section.

Response 1:  We would like to thank the reviewer for the appreciation of our study. As properly commented, one the limitations of this investigation was the use of photographs. However, at the same time, it was an essential requisite to evaluate the intra- and inter-examiner agreement.

Reviewer 2 Report

In this manuscript, the authors tried to evaluate the intra‐ and inter‐examiner reliability of the EHS. Six examiners with different levels of training and clinical focus were enrolled. Each examiner was trained on the use of EHS before starting the study.  The authors described that 63 photographs of three different types of surgical incisions were taken at day 1, 3 or 7 post operatively were independently evaluated according to the proposed assessment method.  A two way random intra‐class correlation coefficient (ICC) and 95% confidence interval (CI) were used to analyze the intra‐ and inter‐examiner reliability for EHS.  They found that the inter‐examiner reliability for EHS was 22 0.828 (95% CI: 0.767 ‐ 0.881). The intra‐examiner reliability ranged between 0.826 (95% CI: 0.728 ‐23 0.891) and 0.915 (95% CI: 0.856 ‐ 0.950). They concluded that the results showed an “almost perfect agreement” for intra‐ and inter‐examiner reliability. EHS provides a system for reproducible repeated ratings for early healing assessment of incisions of periodontal soft tissues. Even when used by examiners with different clinical experience and specialty it shows a high correlation coefficient. It is interesting, however, the sample size and the area for surgery were not described clearly. It is hard to know actually how many cases were evaluated in the studies. The authors described that The sample included photographs of 3 types of surgical incisions (21 vertical releasing incisions; 21 crestal linear incisions; and 21 papilla base incision), taken at 3 post‐operative time points of wound healing (21 from control appointment at 1 day; 21 at 3 days; and 21 at 7 days). The photographs were anonymized regarding patient and operator identity and the nature of surgical procedure performed. For assessment, the final sample of 63 images was assembled in a slideshow and presentation software.  Please identify how many cases were evaluated by each examiner and whether there are any differences between the anterior areas for surgery and posterior area for surgery?

Author Response

Response to Reviewer 2 Comments

We would like to thank the reviewer for his thorough review of the manuscript. We have carefully considered the suggestions and comments from the referee and have implemented requested changes in the revised manuscript, which we feel has thereby improved. In the following, we will respond step-by-step to the concerns raised.

Point 1: It is hard to know actually how many cases were evaluated in the studies. The authors described that The sample included photographs of 3 types of surgical incisions (21 vertical releasing incisions; 21 crestal linear incisions; and 21 papilla base incision), taken at 3 post‐operative time points of wound healing (21 from control appointment at 1 day; 21 at 3 days; and 21 at 7 days). The photographs were anonymized regarding patient and operator identity and the nature of surgical procedure performed. For assessment, the final sample of 63 images was assembled in a slideshow and presentation software.  Please identify how many cases were evaluated by each examiner.

Response 1: The sample size calculation resulted in a minimum of 18 images for 6 examiners to assess EHS reliability. Each participant assessed the inter-examiner and intra-examiner reliability:

for the total inter- and intra-examiner reliability: on the whole sample of 63 images (including all of the three incision types, each of which assessed at day 1, 3 and 7); for the inter-examiner reliability stratified by incisions type on three groups, each of 21 images, divided by incision type (21 vertical releasing incisions; 21 crestal linear incisions; 21 papilla base incisions); Each group of 21 images, referring to a distinct incision type, comprised seven photographs taken at 1, 3 and 7 days, respectively. for the inter-examiner reliability stratified by time points: on three groups, each of 21 images, divided by time points (21 at day 1; 21 at day 3; 21 at day 7). Each group of 21 images, referring to a distinct time point, comprised seven photographs of vertical releasing incisions, seven of crestal linear incisions, and seven of papilla base incisions.

Therefore, we think that the sample size calculation has always been considered and respected, since the reliability assessment – even when stratified by incision type or time points - was made on a minimum of 21 photographs.

In order to avoid any misinterpretation by the reader, we revised the text of the material section accordingly.

Revised Text:

Material and Methods:

Lines 120-127: The sample included photographs of 3 types of surgical incisions (21 vertical releasing incisions, 21 crestal linear incisions and 21 papilla base incision). Each group of 21 images, referring to a distinct incision type, comprised 7 photographs taken at 3 different time points of early wound healing (1, 3 and 7 days). Therefore, for each time point there were 21 images as well. The photographs were anonymized regarding patient and operator identity and the nature of surgical procedure performed.

For assessment, the final composite sample of 63 images was assembled in a slideshow and presentation software. Moreover, a standardized assessment protocol was prepared for data recording.

Line 144: The six examiners independently assessed all 63 images according to EHS indications.

Point 2: there are any differences between the anterior areas for surgery and posterior area for surgery?

Response 2: The authors thank the referee for his critical input. This study mainly focused on the intra-rater and inter-rater agreement between operators with different levels of education and different clinical specialization and how different types of surgical incisions and the time interval at which wounds are assessed could have an influence on the reliability of EHS. However, we considered that it is important to describe the surgical area in which the incisions were localized. At this stage, because it was of questionable interest when evaluating incisions using photographs, we did not calculate the agreement stratified by the surgical area. This aspect could be investigated in further clinical studies. In conclusion, we revised both the Results section and the Discussion section.

Revised Text:

Results:

Lines 185-187: The sample included 39 photographs of incisions performed in the anterior area (from canine to canine) and 26 in the posterior area (from the first premolar to the second molar).  Forty-five of the incisions were localized in the upper arch and 20 lower arch.

Discussion:

Lines 360-366: Inter- and intra-examiner agreement stratified by the surgical area where incisions were localized was not calculated. Even though it could biologically affect the outcome of early wound healing, the anatomical site was not considered significant when evaluating the repeatability of assessments using photographs. Conversely, it would have been important - in terms of agreement - if the evaluation had been carried out clinically. In this case, the access to the surgical area and the operator’s visibility could have been influenced the proper score of early wound healing. Differences in agreement between dissimilar surgical sites could be taken into account in further investigations.

Reviewer 3 Report

The paper “Early wound healing score (EHS): an intra- and inter examiner reliability study.” is an interesting paper analysing the intra‐ and inter‐examiner reliability of the EHS on the basis of the analysis of photographs realized at  different times of healing (1, 3  and 7 days).

I think that the topic is very interestng for clinical researcher due to the difficulties in order to categorize healing stage in a reproductible way.

From my point of view there is no point of wakness in this paper that is well written, easy to read and clear in the analysis.

I think that the proposed method of scoring EWH may be in the future a reference for clinical research on wound healing.

The paper is easy to read and well written.

The analysis is very interesting and the analysis proposed is very useful for surgeon.

Author Response

Point 1: I think that the topic is very interesting for clinical researcher due to the difficulties in order to categorize healing stage in a reproducible way.

From my point of view there is no point of weakness in this paper that is well written, easy to read and clear in the analysis.

I think that the proposed method of scoring EWH may be in the future a reference for clinical research on wound healing.

The paper is easy to read and well written.

The analysis is very interesting and the analysis proposed is very useful for surgeon.

Response 1: We thank the reviewer for the kind appreciation of the work conducted.
